# Effects of Combined Nitrogen–Phosphorus on Biomass Accumulation, Allocation, and Allometric Growth Relationships in *Pinus yunnanensis* Seedlings after Top Pruning

**DOI:** 10.3390/plants13172450

**Published:** 2024-09-02

**Authors:** Guangpeng Tang, Yu Wang, Zhuangyue Lu, Sili Cheng, Zhaoliu Hu, Shi Chen, Lin Chen, Junrong Tang, Yulan Xu, Nianhui Cai

**Affiliations:** 1Key Laboratory of Forest Resources Conservation and Utilization in the Southwest Mountains of China, Ministry of Education, Southwest Forestry University, Kunming 650224, China; tangguangpeng@swfu.edu.cn (G.T.); wangy2711@163.com (Y.W.); luzhuyue@swfu.edu.cn (Z.L.); chengsili@swfu.edu.cn (S.C.); huzhaoliu@swfu.edu.cn (Z.H.); chenshi@swfu.edu.cn (S.C.); linchen@swfu.edu.cn (L.C.); tjrzy2016@swfu.edu.cn (J.T.); xuyulan@swfu.edu.cn (Y.X.); 2Key Laboratory of National Forestry and Grassland Administration on Biodiversity Conservation in Southwest China, Southwest Forestry University, Kunming 650224, China

**Keywords:** *Pinus yunnanensis*, fertilization effect, growth pattern, seedling component

## Abstract

*Pinus yunnanensis* (Franch), a species endemic to southwestern China, provides significant ecological and economic benefits. The quality of afforestation can be enhanced by promoting high-quality sprout growth. This study analyzed the effects of six fertilization treatments following top pruning: T1 (N: 0 g/plant^−1^; P: 0 g/plant^−1^), T2 (N: 0 g/plant^−1^; P: 2 g/plant^−1^), T3 (N: 0 g/plant^−1^; P: 4 g/plant^−1^), T4 (N: 0.6 g/plant^−1^; P: 0 g/plant^−1^), T5 (N: 0.6 g/plant^−1^; P: 2 g/plant^−1^), and T6 (N: 0.6 g/plant^−1^; P: 4 g/plant^−1^). The accumulation and allocation of aboveground biomass in roots, stems, and leaves of *P. yunnanensis* were measured, as well as changes in biomass per plant at 90 days (early stage), 180 days (middle stage), and 270 days (late stage) post-fertilization. At 90 days, root biomass accumulation in T3 was significantly higher, by 13.31%, compared to that in T1 (*p* < 0.05). The growth rates of stem and plant biomass followed the order T6 > T1 > T3 > T5 > T4 > T2. The biomass of sprouts and individual plants exhibited allometric growth under T1, T5, and T6 treatments. At 180 days, needle biomass allocation in T1 and T4 increased by 7.47% and 8.6%, respectively, compared to 90 days. Combined nitrogen–phosphorus fertilization significantly influenced aboveground biomass allocation, promoting growth more effectively than other treatments. By 270 days, the stem and individual biomass in T2 and T3 treatments showed significant differences (*p* < 0.01) compared to other treatments. Overall, root, stem, and sprouts were primarily influenced by phosphorus fertilization, while nitrogen fertilization notably promoted needle and leaf growth in later stages. The aboveground components were more affected by phosphorus fertilization. The combination of nitrogen and phosphorus fertilizers enhanced early-stage stem and sprouts, as well as late-stage root development.

## 1. Introduction

Biomass is a fundamental biological characteristic and functional trait of plants, reflecting their status of material accumulation and ability to utilize environmental resources [1]. The accumulation and allocation of biomass result from, and are necessary drivers of, the comprehensive effects of plant net carbon acquisition [2]. The biomass accumulation characteristics and relative growth patterns of various plant components are primarily determined by genetic factors, but they are also influenced by environmental conditions, management practices, and human interference [3]. The optimal partitioning hypothesis suggests that plants allocate more resources to organs that acquire the most limiting resources, thus enabling the plant to obtain these restricted resources more effectively and maintain the highest growth rate [4]. For instance, under light limitation, biomass is preferentially allocated to leaves and stems, whereas under water and nutrient limitations, it is directed to the roots [5]. Individual size is one of the most critical characteristics of organisms, and the mathematical expression of the relationship between morphological traits and individual size during plant growth constitutes the core content of allometric growth theory [6]. Allometric growth refers to the phenomenon where two traits of an organism grow at different rates, revealing intrinsic, scale-independent relationships between the biomass allocation characteristics of plant components [7]. The allometric partitioning hypothesis posits that resource allocation is influenced by individual size [8]. When individuals are small, plants allocate more resources to leaf production to maximize light capture [6]. However, when significant differences exist in biomass allocation ratios among different organs, the relative growth relationships between organs may not be evident. Simply studying biomass allocation between organs may not accurately reflect the plant’s balanced strategy for supplying environmental resources [9]. Therefore, combining the analysis of biomass allocation with relative growth relationships can better elucidate the biomass allocation characteristics of plants [10].

Pine trees, known for their strong adaptability [11], are widely distributed across China. *Pinus yunnanensis* (Franch), a species endemic to southwestern China, is a pioneer species in the natural regeneration of barren mountains in the Yunnan-Guizhou Plateau. This species not only conserves soil and water, prevents wind erosion, and stabilizes sand but is also utilized for resin production, papermaking, tannin extraction, and medicine [12], thereby providing significant ecological and economic benefits. However, the low productivity of *P. yunnanensis* plantations hinders sustainable development. High-quality sprouts are crucial for the restoration and ecological construction of *P. yunnanensis*, which can be achieved through top pruning to promote seedling sprouting, thereby yielding high-quality scions. Studies have shown that pruning seedlings can increase uniformity [13], promote growth, and enhance survival rates [14]. Top pruning effectively stimulates the growth of adventitious buds at the base of the stem, rapidly forming a forest canopy with nutrient support from the roots [15]. Additionally, top pruning improves light conditions within the plant population, reduces leaf shading, enhances light energy utilization efficiency, and redistributes nutrients among various organs, thereby increasing plant yield [16]. Following top pruning, a significant portion of the water absorbed by the root system is utilized for sprout growth and recovery. During this recovery phase, plants adjust their morphological structure, growth rate, and biomass allocation to adapt to changing moisture conditions [17]. However, due to insufficient soil fertility, the resulting scions often fail to meet expected quality standards. Therefore, it is imperative to investigate the biomass changes and allocation patterns of various components in *P. yunnanensis* seedlings under nitrogen and phosphorus fertilization conditions after top pruning. Such research will contribute to the development of high-quality scions and enhance the ecological and economic benefits of this species.

Studies have reported that abundant nutrient reserves can promote seedling growth and enhance drought resistance [18]. Nitrogen (N) and phosphorus (P) are essential nutrients for plant growth and development, as they constitute key organic compounds and play crucial roles in various physiological and metabolic processes [19,20]. These nutrients significantly influence plant productivity [21], photosynthesis rates [22], and other ecosystem functions [23]. Nitrogen and phosphorus are often limiting factors in plant biomass production, making them critical to the structure, processes, and functions of ecosystems [24,25]. Nitrogen input has been shown to positively impact plant productivity and facilitate vegetation restoration [26,27]; for example, nitrogen fertilizer application can increase growth by increasing leaf area, branch size, and branch longevity [28], as well as reducing the proportion of biomass partitioned below ground [29]. Phosphorus, the second most important nutrient after nitrogen, is considered one of the most limiting factors for sapling growth during the initial production phase due to its low availability in the soil [30]. Phosphorus deficiency can lead to fewer leaves and lateral roots, as well as a reduction in the biomass production of the aerial parts of forest species in commercial plantations [31,32]. Proper fertilization practices can enhance forest growth [33] and optimize the nutrient content within the forest ecosystem [34]. The benefits of fertilization can be observed through increased biomass and improved allometric growth [21]. This study utilized one-year-old *P. yunnanensis* seedlings subjected to top pruning to analyze the changes in biomass and allometric growth relationships among roots, stems, needles, and sprouts under varying nitrogen and phosphorus fertilization regimes over time. The objective was to investigate the spatial and temporal patterns of biomass accumulation, allocation, and allometric growth strategies in response to nitrogen and phosphorus additions. The findings aim to provide insights for top pruning fertilization, forest regeneration, and ecosystem restoration.

## 2. Results

### 2.1. Effect of Combined Nitrogen–Phosphorus on Root Biomass Accumulation and Allocation

At 90 and 180 days after fertilization, a single phosphorus application significantly affected root biomass accumulation (Table 1), but the allocation was affected by combined nitrogen–phosphorus (Table 2). The growth of seedlings treated with fertilization was higher than that without fertilization (T1), among which T3 showed the best effect (*p* < 0.05), which was significantly different from that on T1 (Figure 1A). With the growth of seedlings, at 270 days, the biomass allocation showed T3 > T4 > T5 > T6 > T1 > T2, but there was no significant difference (Figure 2A). At 90 days, root biomass accumulation in T3 (27.2%) was 13.31% higher than in T1 (13.89%).

### 2.2. Effects of Combined Nitrogen–Phosphorus on Stem Biomass Accumulation and Allocation

At 90 days after fertilization, the application of phosphate fertilizer alone significantly promoted stem growth. By 270 days, stem biomass accumulation was predominantly influenced by phosphate fertilizer (Table 1), while the allocation of stem biomass was significantly impacted by nitrogen fertilizer (Table 2), with highly significant differences observed (*p* < 0.01). Over the period from 90 to 270 days, stem biomass accumulation was mainly affected by T2 and T3, followed by T5 and T6 (Figure 1B). Stem biomass remained relatively stable during the early and middle stages, with significant differences emerging between T2 and T4, and T5 and T6 in the later stage (Figure 2B).

### 2.3. Effects of Combined Nitrogen–Phosphorus on Needle Biomass Accumulation and Allocation

In terms of needle biomass accumulation and allocation at 90 days, the performance followed the order phosphorus fertilizer treatment > nitrogen fertilizer treatment > combined nitrogen–phosphorus treatment (Table 1 and Table 2). Specifically, the biomass allocation ratios for T1 and T4 at 90 days were 6.56% and 17.18%, respectively. By 180 days, these ratios increased to 15.16% and 24.65%, representing increases of 8.6% and 7.47%, respectively. At 180 days, T2 achieved the highest biomass, while T4 had the lowest (Figure 1C). Compared to 90 days, biomass allocation in T1 and T4 increased, although the allocation ratio of needles gradually decreased as the seedlings continued to grow (Figure 2C).

### 2.4. Effects of Combined Nitrogen–Phosphorus on Sprout Biomass Accumulation and Allocation

By analyzing the sources of biomass accumulation and allocation variation, it was concluded that phosphate fertilizer treatment exhibited the most effective growth, with significant differences from other treatments at both 180 and 270 days (Table 1). As shown in Figure 1D, with the extension of time, biomass accumulation during the later stages of fertilization followed the order T2 > T3 > T1 > T5 > T4 > T6. In the early stages, biomass allocation in sprouts was primarily influenced by phosphorus fertilizer, while in the later stages, nitrogen fertilizer played a more significant role (Table 2). Fertilization increased the allocation ratio with seedling growth (Figure 2D), with T2 and T3 showing the most effective allocation.

### 2.5. Effects of Combined Nitrogen–Phosphorus on Aboveground Biomass Accumulation and Allocation

Aboveground biomass, composed of stem, needle, and sprout biomass, showed significant differences under the combined nitrogen–phosphorus treatment at 90 days when compared with other treatments (Table 2). As fertilization time increased, T2, T3, and T5 exhibited a strong promoting effect on the growth of aboveground components (Figure 1E). From 90 to 270 days, biomass allocation showed an initial increase followed by a decrease. Notably, T5 fertilization significantly reduced aboveground biomass from 180 to 270 days (Figure 2E).

### 2.6. Effects of Combined Nitrogen–Phosphorus on Biomass Accumulation Individual 

As shown in Figure 1F, individual plant biomass accumulation was more pronounced under T2, T3, and T5 fertilization. Generally, except for T5 and T6, there was no significant growth between 180 and 270 days; however, biomass under other treatments increased significantly over time.

### 2.7. The Relationship between Biomass Accumulation and Allocation of Each Component with Time under Different Fertilization Treatments

Figure 3 illustrates that fertilization timing has a highly significant positive correlation with biomass accumulation across all components of *P. yunnanensis*. As seedlings grew, there was a significant positive correlation between root and sprout biomass allocation, and a significant negative correlation with needle biomass allocation. Aboveground biomass accumulation was positively correlated with the biomass accumulation of roots, stems, needles, and sprouts. However, the biomass allocation of stems and needles was inversely correlated with sprout biomass accumulation. Sprout biomass allocation, needle biomass accumulation, and allocation were all negatively correlated.

### 2.8. Allometric Growth Relationships of Different Components and Individual Plant Biomass under Combined Nitrogen–Phosphorus at Different Stages

As fertilization time progressed from 90 to 270 days, root-individual biomass (Figure 4A, Figure 5A and Figure 6A) showed faster growth in T1 during the early stages, with isometric growth relationships observed in the middle and later stages. In contrast, T6 exhibited faster growth in the later stages, displaying an allometric growth relationship. For stem-individual biomass (Figure 4B, Figure 5B and Figure 6B), T1 and T3 showed isometric growth as fertilization time increased in the later stages. Regarding leaf-individual biomass (Figure 4C, Figure 5C and Figure 6C), T4 effectively promoted needle growth at 270 days, while T2 had no noticeable effect, indicating isokinetic growth. In terms of sprout-individual biomass (Figure 4D, Figure 5D and Figure 6D), T3 had a higher growth rate, while T1 exhibited slower growth during the early and middle stages as seedlings matured. For aboveground-individual biomass (Figure 4E, Figure 5E and Figure 6E), T1 and T2 showed allometric growth in the early and middle stages, transitioning to isokinetic growth in the later stages as fertilization time extended. Lastly, for aboveground and underground biomass (Figure 4F, Figure 5F and Figure 6F), the growth rate of T4 seedlings was higher in the later stages compared to the early and middle stages, showing an allometric growth pattern.

## 3. Discussion

### 3.1. Changes in Biomass Accumulation and Allocation of Various Components

Several studies have confirmed the effects of fertilization on tree growth and other properties related to productivity. Studies have shown that both annual fertilization and one-time applications result in increased productivity in southern pines [35,36]. In this study, we found that there were significant differences (*p* < 0.05) in biomass accumulation and allocation of different fractions (roots, stems, leaves, and shoots) under different treatments at different periods. This indicates that fertilization increases individual biomass and promotes growth [37]. Studies have shown that nitrogen and phosphorus fertilizers alter nutrient uptake and partitioning in the aboveground and below-ground portions of the tea tree. Consequently, biomass accumulation and allocation are different [38]. In this study, the combined application of nitrogen and phosphorus primarily increased root biomass allocation in the middle and late stages of growth. Aboveground biomass was predominantly accumulated during the early and middle stages, with higher biomass allocation observed in the later stages under non-fertilized conditions. According to the theory of allometric biomass allocation [39], larger plants tend to allocate more resources to the main stem and roots. In this study, aboveground biomass accumulation and allocation were greater than those of belowground parts, indicating a preferential allocation of resources to aboveground components. As seedling growth progressed, needle biomass accumulation and allocation gradually decreased, while root and stem biomass showed an upward trend. 

Fertilization treatments in this study resulted in greater biomass accumulation compared to the control, with phosphorus fertilizer effectively promoting root and stem biomass accumulation, and the combined application of nitrogen and phosphorus significantly influencing needle allocation. The biomass accumulation effect of high-concentration phosphate fertilizer was more pronounced in roots than in stems, where low-concentration phosphate fertilizer had a greater effect. This suggests that fertilization altered the nutritional environment for seedling growth, with seedlings responding sensitively to changes in nutrient supply by adjusting their physiological and morphological traits to adapt to environmental conditions [40,41]. Among these, nitrogen and phosphorus fertilization improved seedling quality and stress resistance by promoting growth and biomass accumulation [42,43], although the type and concentration of fertilizers could restrict seedling growth states and rates. From 90 to 270 days, root biomass accumulation in the control group showed a continuous significant increase, whereas in fertilized treatments, it initially increased steadily before showing a significant rise. Needle biomass accumulation in fertilized treatments displayed a steady continuous increase, while in the control group, it first increased steadily and then significantly. Biomass allocation in the control group continuously increased steadily, while in fertilized treatments, it showed a significant increase between 180 and 270 days. Aboveground biomass allocation in the control group initially increased and then decreased without significant differences; in fertilized treatments, it initially increased steadily and then decreased significantly. Fertilization treatments modified biomass accumulation and allocation patterns over time (Figure 1 and Figure 2). These treatments resulted in better allocation than in the control, suggesting that the combined application of nitrogen and phosphorus effectively promotes biomass accumulation and allocation following top pruning, consistent with the findings of Dovrat et al. [44]. In treatments T1, T2, and T3, needle biomass allocation and accumulation initially increased and then decreased over time, likely due to the long-term lack of nitrogen regulation and synthesis, which reduced needle biomass accumulation and allocation. Some studies suggest that seedlings gradually recover growth through sprouting after aboveground loss, balancing growth, reproduction, and defense functions with limited resource allocation [45]. In this study, sprout biomass constituted the largest allocation, followed by roots and stems, with needles receiving the least allocation. This pattern suggests that seedlings prioritize sprout growth to support photosynthesis and maintain vital activities, reflecting an adaptive strategy to external environmental changes.

### 3.2. Allometric Growth Relationships of Various Components and Individual Plant Biomass

In this study, from 90 to 270 days, the unfertilized treatment relatively promoted root growth, resulting in allometric growth, with less allocation to the aboveground parts, while the fertilized treatments enhanced aboveground growth, leading to reduced root allocation. The results demonstrate that variations in organism size and their components reflect differences in growth rates at different stages, with distinct organs exhibiting varying growth effects over time, a phenomenon known as allometric growth. This highlights the quantitative relationship between growth and allocation, which has profound implications for the utilization, growth, development, and reproduction of biological resources [46,47,48,49]. Biomass allocation and allometric growth typically vary with species and nutrient availability [50]. Under low-nutrient conditions, plants preferentially allocate resources to the roots to enhance essential nutrient uptake [51]. In this study, from 90 to 270 days, the unfertilized treatment relatively promoted root growth, resulting in allometric growth with less allocation to aboveground parts, while the fertilized treatments promoted aboveground growth, leading to reduced allocation to roots.

At 90 days, the growth trends of roots, stems, needles, sprouts, aboveground biomass, and individual plant biomass varied, altering their relative growth relationships. Stems, needles, and belowground biomass in the T4 and T5 treatments exhibited isometric growth compared to the unfertilized treatment. Roots and sprouts in the T2 and T3 treatments also demonstrated isometric growth, indicating that fertilization increased the biomass of all components while also enhancing individual plant biomass, whereas the individual plant biomass in the unfertilized treatment was lower. Consequently, the components and individual plant biomass in the unfertilized treatment exhibited allometric growth. The growth relationship between aboveground and belowground biomass in fertilized and unfertilized treatments was similar, suggesting that fertilization did not alter their relative growth relationships. At 180 days, the growth relationship between roots and stems was not apparent in both fertilized and unfertilized treatments, with no allometric growth observed. In the needles, T1, T5, and T6 treatments showed allometric growth, with the growth rate of nitrogen–phosphorus combined fertilization being higher than that of the unfertilized treatment. Previous studies have demonstrated that nitrogen positively influences photosynthesis rates [52]. In this study, the growth rate of sprouts in the single nitrogen fertilization treatment increased, promoting sprout growth, consistent with these findings. Regarding the relationship between aboveground and individual biomass, overall growth was stable in the fertilized treatments, displaying isometric growth. At 270 days, root growth rates in the T4 and T6 fertilization treatments were higher than in the unfertilized treatment, but fertilization did not alter the relative growth relationship in stems, maintaining isometric growth. Sprouts in the T3 and T6 treatments exhibited the best growth, effectively promoting growth, indicating that sprout metabolism was primarily regulated by phosphorus, with growth increasing with phosphorus content, consistent with previous reports [53]. In the relationship between aboveground and individual biomass, fertilization did not alter growth rates, while the growth rate of aboveground and belowground biomass in the T4 treatment increased, indicating a higher nitrogen demand for photosynthesis in the aboveground parts compared to the belowground parts. This study found that (1) fertilization after top pruning effectively promoted the growth of *P. yunnanensis* seedlings and increased biomass, consistent with the results of previous studies [35,36]. (2) The high proportion of sprout biomass indicates that seedlings prioritize sprout growth to ensure photosynthesis and life activities, reflecting an adaptive strategy to cope with environmental resource limitations [54,55,56].

Previous studies have shown [6] that smaller plants prioritize needle allocation. In this study, early aboveground biomass and individual plant size in T1 and T6 exhibited significant allometric growth, indicating that combined nitrogen–phosphorus fertilization effectively promoted aboveground growth, enhancing photosynthetic capacity. In the mid-stage, fertilization did not significantly affect the growth of roots and stems, nor alter their growth rate relationships. However, it significantly promoted needle and sprout growth, with faster growth rates, indicating that more resources were allocated to needles and sprouts during the mid-stage, resulting in lower growth rates of roots and stems. As plants grew larger, resource allocation to leaves decreased [57]. In this study, as seedlings matured, individual plants increased in size. In the later stages, root biomass gradually increased, with higher resource allocation, showing allometric growth in the T4 and T6 treatments, consistent with this theory. Overall, the effects of combined fertilization were greater than those of single fertilization on growth rates. Combined nitrogen–phosphorus fertilization increased stem biomass accumulation, while high-quality single phosphorus fertilization promoted root growth. Single nitrogen fertilization resulted in higher growth rates in stems, indicating that nitrogen fertilization promoted stem and branch growth, while phosphorus fertilization enhanced root development.

Post-top pruning, fertilization can improve seedling survival rates, promote production and development, and increase biomass. These results are consistent with the findings of Jokela and Borders [35,36]. The high proportion of sprout biomass suggests that seedlings prioritize sprout growth to ensure photosynthesis and external activities. This reflects an adaptive strategy to respond to external environmental changes and manage environmental resources [54,55,56].

## 4. Materials and Methods

### 4.1. Study Area

The experiment was conducted at the nursery of Southwest Forestry University, located at approximately 102°45′41″ E, 25°04′00″ N, with an elevation of about 1945 m. The area has an average annual temperature of 14.7 °C, an absolute minimum temperature of −9 °C, and an absolute maximum temperature of 32.5 °C. The annual precipitation ranges from 700 to 1100 mm, with an average relative humidity of 68.2%. It falls under the category of a semi-humid plateau monsoon climate in the northern subtropics, with acidic, low-phosphorus soil (pH = 6.0~6.2, total phosphorus conten: 0.9~1.22 g/kg).

Seeds for the experiment were obtained from the *P. yunnanensis* clonal seed orchard in Midu County, Yunnan Province. Seeds were collected from healthy mother trees bearing mature cones of the current year. After proper labeling and drying, well-developed seeds were selected. The seeds were uniformly disinfected and soaked, followed by sowing in May 2021. After complete germination, seedlings were transplanted in August to nursery pots (dimensions: bottom diameter 18 cm × height 32 cm) filled with a substrate mixture of humus soil and red soil in a 3:1 ratio. Seedlings of uniform size were selected for the subsequent experiments and were uniformly managed throughout the experimental period, with weeding every 2 weeks and watering every 3 to 5 days until saturation.

### 4.2. Experimental Design

The experiment involved the combination of two ingredients: N and P. N fertilizer used was urea with a total N content of 46.40%, applied at rates of 0.0 g/plant and 0.6 g/plant. P fertilizer used was calcium superphosphate with a P pentoxide (P_2_O_5_) content of 85%, applied at rates of 0.0 g/plant, 2.0 g/plant, and 4.0 g/plant. This resulted in a total of 6 treatments (see Table 3). Each treatment had 35 seedlings with 3 replicates, totaling 18 plots and seedlings. In February 2022, all seedlings were uniformly top pruned at 5 cm. In March, according to the experimental design, different nitrogen–phosphorus ratios were thoroughly mixed with the substrate for the first application, followed by a second application in June (same as the March application).

### 4.3. Biomass Measurement

After experimental setup, they were collected at 90, 180, and 270 days post-fertilization. In each treatment, 3 seedlings were randomly selected with three replicates, totaling 54 seedlings, for biomass measurement. The method was as follows: using the whole excavation method, entire seedlings were obtained and placed in labeled bags for transportation to the laboratory. The collected samples were washed with clean water, drained, and divided into four parts: roots, stems, needles, and sprouts. After drying in an oven at 105 °C for 30 min to deactivate enzymes, they were further dried at 80 °C until a constant weight was achieved, and the dry biomass was recorded (accurate to 0.0001 g). The biomass allocation ratio was calculated as follows: biomass of a component/individual biomass × 100% (e.g., root biomass allocation ratio = root biomass/individual biomass × 100%). Individual biomass was calculated as the sum of root biomass and aboveground biomass, where aboveground biomass was the sum of stem, needle, and sprout biomass.

### 4.4. Data Analysis

Data collected were organized using Excel 2020 and statistically analyzed using SPSS 20.0. One-way analysis of variance (ANOVA) was performed for the 6 different treatments at 90, 180, and 270 days post-top pruning, followed by multiple comparisons using Duncan’s method. Two-way ANOVA was conducted to examine the effects of N, P, and combined nitrogen–phosphorus on biomass allocation in seedlings post-top pruning. The allometric growth relationship can be expressed as $y=ax^{b}\;$ [56], which linearizes to logy = loga + blogx. Here, y and x represent the biomass of individual seedlings and the biomass of various components (roots, stems, needles, sprouts, and aboveground biomass). Parameters were estimated using the standard major axis method, and comparisons of allometric growth indices and intercepts were conducted using the statistical software package R V2.0, specifically the Smatr package [58,59]. Data are presented as mean ± standard error, and graphs were generated using Origin 2021 and GraphPad Prism 9.0.

## 5. Conclusions

Prolonged fertilization can enhance the biomass accumulation and allocation in various seedling components, with most components showing greater biomass accumulation and allocation after fertilization compared to non-fertilized conditions. Phosphorus fertilizer effectively promotes the growth of roots, stems, and sprouts in seedlings, while nitrogen fertilizer has a significant effect on needle growth at later stages. The aboveground parts are primarily influenced by phosphorus fertilizer and the combined application of nitrogen and phosphorus. Both nitrogen and phosphorus fertilizers can improve the sprouting ability of seedlings, with phosphorus fertilizer showing the most pronounced effect. Fertilization also alters the allocation pattern among different components. Specifically, needle biomass allocation decreases with the increase in sprout growth. Stem allocation is constrained by nitrogen, and the combined application of nitrogen and phosphorus. The allocation ratio of aboveground biomass reaches its peak at 180 days post-fertilization, and the combined application of nitrogen and phosphorus further enhances this allocation. Biomass accumulation and allocation during different growth periods were generally higher in aboveground parts than in belowground parts, and higher in sprouts compared to roots, stems, and needles. In conclusion, it is recommended to apply a combined nitrogen–phosphorus treatment following seedling top pruning to promote recovery and enhance growth. Future research should focus on the following aspects: (1) utilizing molecular biotechnology to investigate the genetic mechanisms underlying the combined nitrogen–phosphorus treatment’s role in promoting seedling recovery; (2) exploring the effects of combined nitrogen–phosphorus application on stress resistance (e.g., drought resistance, cold tolerance, disease, and pest resistance) in *P. yunnanensis* seedlings and the associated mechanisms.

## Figures and Tables

**Figure 1 plants-13-02450-f001:**
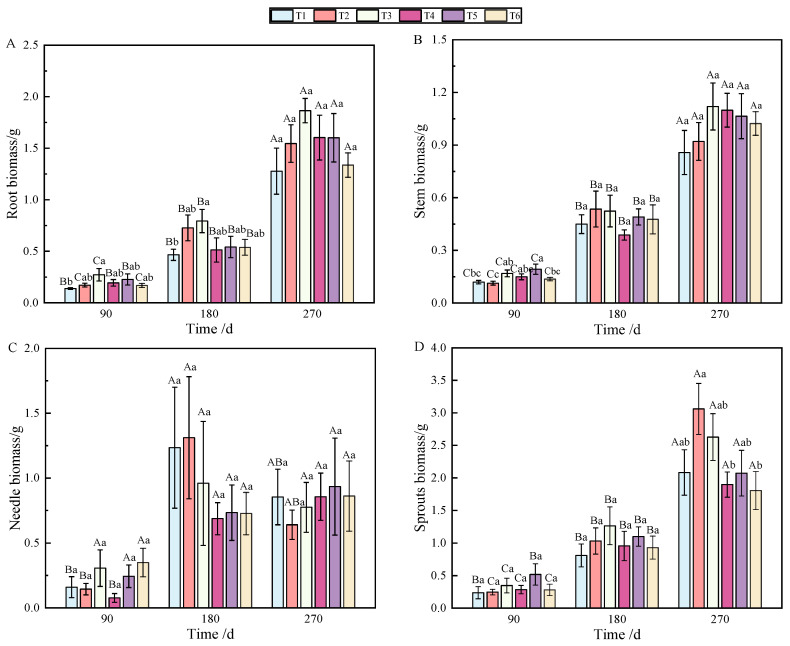
Changes in biomass accumulation of roots (**A**), stems (**B**), needles (**C**), sprouts (**D**), aboveground parts (**E**), and individuals (**F**) of *P. yunnanensis* with fertilization time. (Note: different uppercase letters indicate significant differences between times within the same treatment (*p* < 0.05). Different lowercase letters indicate significant differences between treatments at the same time (*p* < 0.05); the same as below).

**Figure 2 plants-13-02450-f002:**
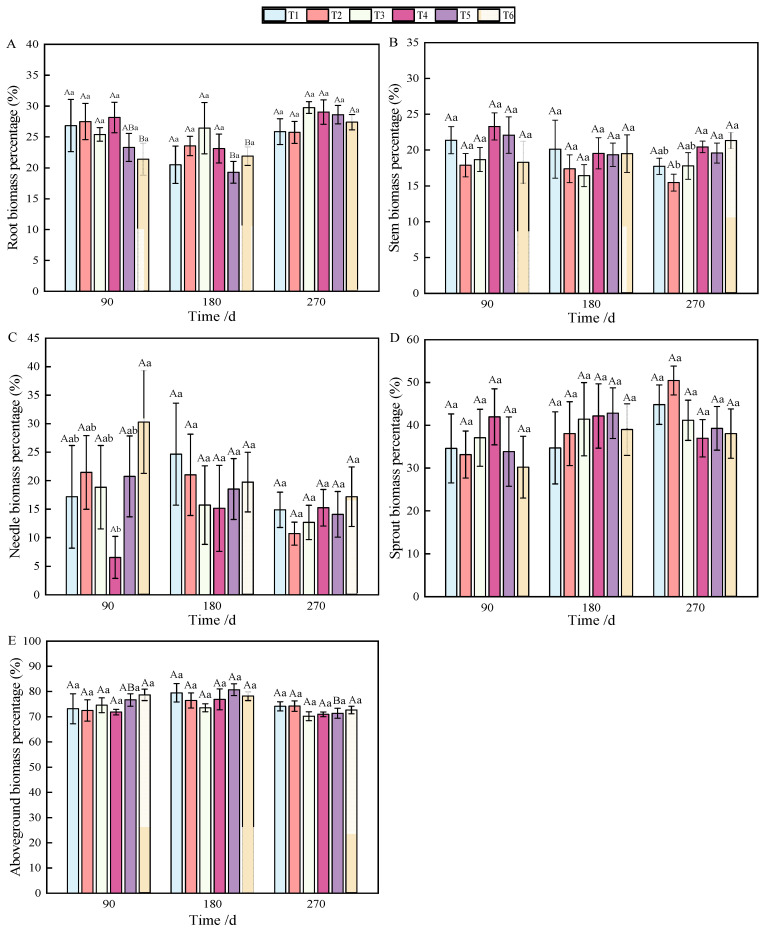
Changes in biomass allocation of roots (**A**), stems (**B**), needles (**C**), sprouts (**D**), and aboveground parts (**E**) of *P. yunnanensis* with fertilization time.

**Figure 3 plants-13-02450-f003:**
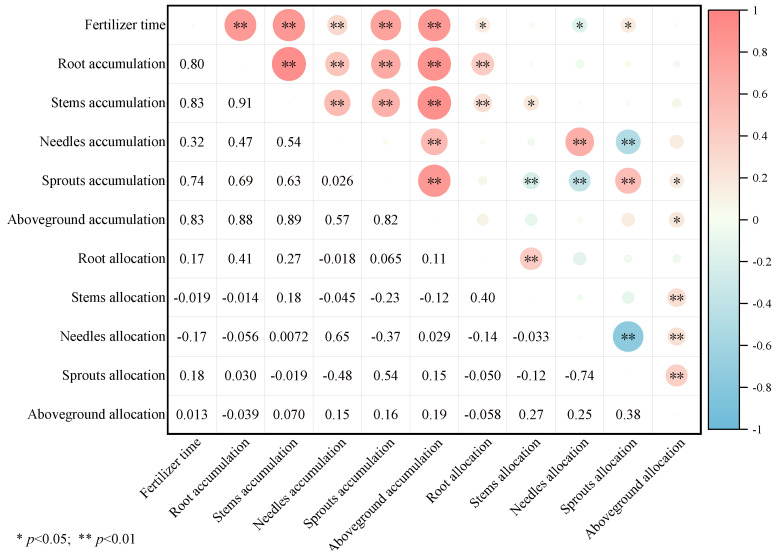
The correlation between biomass accumulation and distribution of each component of *P. yunnanensis* with time under different fertilization treatments. (Note: * *p* < 0.05; ** *p* < 0.01).

**Figure 4 plants-13-02450-f004:**
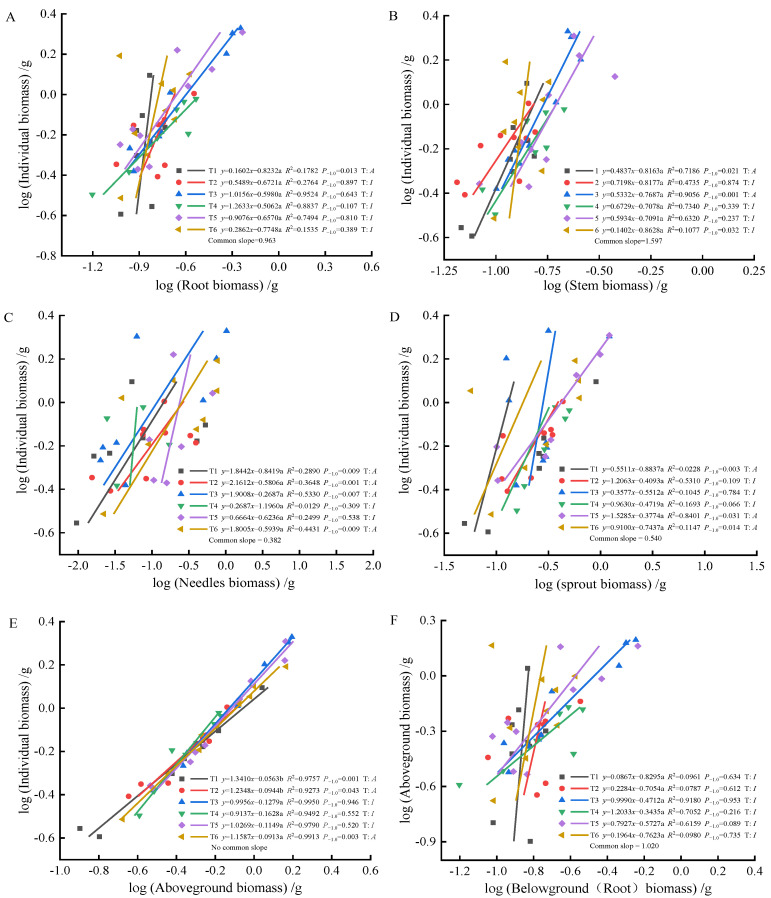
At 90 days, the biomass of individual plant and root (**A**), stem (**B**), needle (**C**), sprout (**D**), aboveground biomass (**E**); the allometric relationship between aboveground and belowground biomass (**F**). (Note: P_−1.0_ indicates the significance of the difference between the slope and the theoretical value of 1.0. ‘T’ indicates the type, ‘A’ indicates allometric relationships, ‘I’ indicates isometric relationships).

**Figure 5 plants-13-02450-f005:**
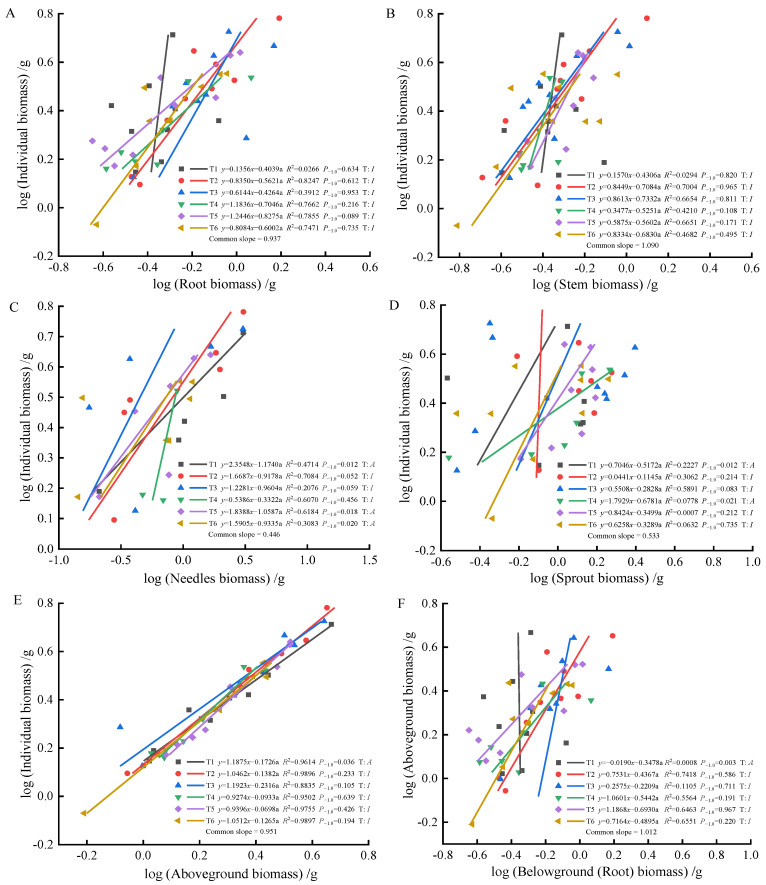
At 180 days, the biomass of individual plant and root (**A**), stem (**B**), needle (**C**), sprout (**D**), aboveground biomass (**E**), the allometric relationship between aboveground and belowground biomass (**F**).

**Figure 6 plants-13-02450-f006:**
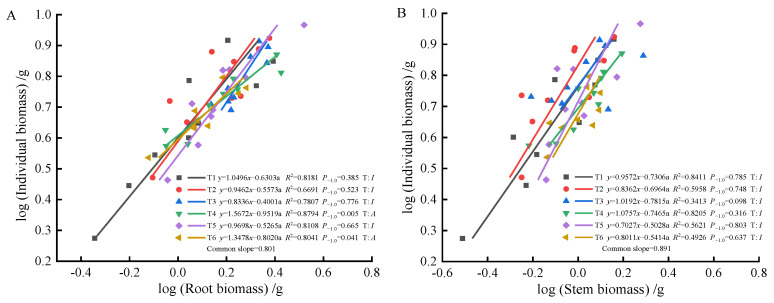
At 270 days, the biomass of individual plant and root (**A**), stem (**B**), needle (**C**), sprout (**D**), aboveground biomass (**E**), the allometric relationship between aboveground and belowground biomass (**F**).

**Table 1 plants-13-02450-t001:** Analysis of variation sources of biomass accumulation in each component of *P. yunnanensis* seedlings after top pruning.

Time/d		90				180			270		
Source of Variation		df	SS	MS	F	SS	MS	F	SS	MS	F
root	N	1	0.000	0.000	0.007	0.272	0.272	3.645	0.032	0.032	0.754
	P	2	0.105	0.052	4.267 *	1.024	0.512	6.853 **	1.458	0.729	0.116
	N × P	2	0.007	0.003	0.266	0.135	0.067	0.900	0.360	0.180	0.577
stem	N	1	0.007	0.007	2.703	0.035	0.035	0.913	0.125	0.125	1.472
	P	2	0.019	0.009	3.443 *	0.376	0.188	4.917 *	1.555	0.778	9.192 **
	N × P	2	0.011	0.005	2.003	0.182	0.091	2.374	0.197	0.098	1.162
needle	N	1	0.002	0.002	0.352	1.951	1.951	2.530	0.316	0.316	0.13
	P	2	0.451	0.226	3.582 *	0.359	0.180	0.233	1.782	0.891	2.578
	N × P	2	0.024	0.012	0.189	0.102	0.102	0.132	1.959	0.979	2.834
sprout	N	1	0.072	0.072	0.830	0.053	0.053	0.159	5.981	5.981	6.416 *
	P	2	0.100	0.05	0.578	2.648	1.324	3.977 *	0.267	0.134	0.143
	N × P	2	0.019	0.010	0.111	0.252	0.126	0.378	6.514	3.257	3.494 *
aboveground	N	1	0.092	0.092	0.717	2.060	2.060	2.779	2.529	2.529	2.083
	P	2	0.571	0.286	2.232	10.289	5.144	6.939 **	7.087	3.543	2.918
	N × P	2	0.235	0.117	0.916	1.858	0.929	1.253	3.026	1.513	1.246

(Note: N represents single nitrogen application; P represents single phosphorus application; N × P represents a combined application of N and P; SS and MS respectively represent the sum of squares and mean square, * *p* < 0.05; ** *p* < 0.01; the same as below).

**Table 2 plants-13-02450-t002:** Analysis of variance in biomass allocation ratios of various components of *P. yunnanensis* seedlings after top pruning.

Time/d		90				180			270		
Source of Variation		df	SS	MS	F	SS	MS	F	SS	MS	F
root	N	1	0.011	0.011	1.814	0.008	0.008	1.369	0.002	0.002	0.759
	P	2	0.002	0.001	0.190	0.009	0.004	0.798	0.001	0.000	0.191
	N × P	2	0.054	0.027	4.520 *	0.025	0.013	2.275	0.000	0.000	0.033
stem	N	1	0.003	0.003	0.834	0.003	0.003	0.640	0.016	0.016	12.307 **
	P	2	0.012	0.006	1.634	0.016	0.008	1.551	0.011	0.005	4.040
	N × P	2	0.015	0.008	1.993	0.011	0.005	1.079	0.003	0.001	1.022
needles	N	1	0.002	0.002	0.044	0.004	0.004	0.155	0.012	0.012	1.660
	P	2	0.436	0.218	4.900 *	0.698	0.349	13.215 **	0.030	0.015	1.989
	N × P	2	0.018	0.009	0.202	0.050	0.025	0.950	0.053	0.026	3.520 *
sprout	N	1	0.000	0.000	0.000	0.009	0.009	0.273	0.074	0.074	4.128 *
	P	2	0.314	0.157	4.152 *	0.620	0.310	9.414 **	0.057	0.029	1.604
	N × P	2	0.041	0.021	0.548	0.062	0.031	0.938	0.082	0.041	2.282
aboveground	N	1	0.011	0.011	1.814	0.008	0.008	1.369	0.002	0.002	0.759
	P	2	0.002	0.001	0.190	0.009	0.0041	0.798	0.001	0.000	0.191
	N × P	2	0.054	0.027	4.520 *	0.025	0.013	2.275	0.000	0.000	0.033

(* *p* < 0.05; ** *p* < 0.01).

**Table 3 plants-13-02450-t003:** Experimental design for N and P addition in *P. yunnanensis* seedlings.

Treatment	N/(g/plant^−1^)	P/(g/plant^−1^)
N1P1 (TI)	0.0	0.0
N1P2 (T2)	0.0	2.0
N1P3 (T3)	0.0	4.0
N2P1 (T4)	0.6	0.0
N2P2 (T5)	0.6	2.0
N2P3 (T6)	0.6	4.0

## Data Availability

All data generated or analyzed during this study are included in this article. All data are available upon reasonable request.

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
