# Peer review of "Effects of Combined Nitrogen–Phosphorus on Biomass Accumulation, Allocation, and Allometric Growth Relationships in Pinus yunnanensis Seedlings after Top Pruning"

_plants, 2024, doi:10.3390/plants13172450_

Round 1

Reviewer 1 Report

Comments and Suggestions for Authors

see attached

Comments on the Quality of English Language

Papers is generally well written; there are some errors and some confusing sections and these have been noted.

Author Response

Dear Editors:

On behalf of my co-authors, we thank you very much for allowing us to revise our manuscript; we appreciate the editor and reviewers very much for their positive and constructive comments and suggestions on our manuscript entitled “Effects of Combined Nitrogen-Phosphorus on Biomass Accumulation, Allocation, and Allometric Growth Relationships in Pinus yunnanensis Seedlings After Top Pruning.”

We have studied reviewer's comments carefully and have made revision which marked in red in the paper. We have tried our best to revise our manuscript according to the comments.Attached please find the revised version, which we would like to submit for your kind consideration.

We would like to express our great appreciation to you and reviewers for comments on our paper. Looking forward to hearing from you.

Thank you and best regards.

Reviewer 2 Report

Comments and Suggestions for Authors

Very extensive research and interesting results. However, percentage changes should be provided in the abstract and description of results. In the introduction and discussion, provide the plant species to which the individual cited publications refer. In the discussion, herbaceous plants (rice, cotton, potatoes) should be replaced with woody plants because this is what the study was about. The physiology of these groups of plants is different, and here the uptake and accumulation of the given ingredients in plant organs was examined throughout the entire growth cycle.

Comments on the Quality of English Language

Sufficient level.

Author Response

(The authors gave the same response as above.)

Round 2

Reviewer 1 Report

Comments and Suggestions for Authors

This manuscript has been greatly improved. I would suggest that the editor have someone read through it carefully catching any small writing errors.  As an unpaid, 80-year-old retiree, it is not my job to be an editor!

My only comment comes after I read the manuscript once again and I found my self stumbling over the T1, T2, etc. indicators of the six different treatments (this was especially true for the abstract where a reader would have absolutely no idea what the Ts stood for and thus saying T2 is equal to or greater or less than T3 is meaningless.  I thought of a new way of doing T1 through T6.  T1 is C for control, T2 = N+, T3 = N++, T4 = P, T5 = PN+, and T6 = PN++.  I realize this makes a lot of work as text, tables and figures need changes; however, find and replace is a fast function.  I think this would help readers greatly.  If they cannot understand or easily understand the abstract, why would they even attempt to read the rest.

Comments on the Quality of English Language

Much improved

Author Response

Comments 1: [My only comment comes after I read the manuscript once again and I found my self stumbling over the T1, T2, etc. indicators of the six different treatments (this was especially true for the abstract where a reader would have absolutely no idea what the Ts stood for and thus saying T2 is equal to or greater or less than T3 is meaningless.  I thought of a new way of doing T1 through T6.  T1 is C for control, T2 = N+, T3 = N++, T4 = P, T5 = PN+, and T6 = PN++.  I realize this makes a lot of work as text, tables and figures need changes; however, find and replace is a fast function.  I think this would help readers greatly.  If they cannot understand or easily understand the abstract, why would they even attempt to read the rest.]

Response 1: [Your comments I carefully read and carefully modified, combined with your advice, I will each deal with the specific amount of fertilizer, written in brackets after the corresponding treatment. For example, T1 was modified to T1 (N: 0g / plant-1 ; P: 0g / plant-1), T2 was modified to T2 (N: 0g / plant-1 ; P: 2g / plant-1), etc. This can make readers better read and understand, the specific lines 19 to 21.] Thank you for pointing this out.

Reviewer 2 Report

Comments and Suggestions for Authors

Despite the small number of my comments, the authors of the publication still did not take them into account, see introduction. In the results, provide the values ​​or percentage significance of changes in the description of the results. In the discussion, the authors continue to use annual plants, e.g. potatoes and rice, and as a result, there has been a slight change in the literature list. The lack of changes requested by the reviewer was not explained.

Comments on the Quality of English Language

Sufficient level.

Author Response

Comments 1: [In the results, provide the values ​​or percentage significance of changes in the description of the results.]

Response 1: [Your suggestion I have carefully read and carefully modified, the results appear in the value or percentage, are the difference × 100 %.] Thank you for pointing this out.

Comments 2: [In the discussion, the authors continue to use annual plants, e.g. potatoes and rice, and as a result, there has been a slight change in the literature list. The lack of changes requested by the reviewer was not explained. The specific data are obtained from the corresponding columnar values in Figure 1 and Figure 2, so the corresponding data tables or graphs are not repeated separately.]

Response 2: [References related to annual plants such as soybeans, rice, barley, and cotton appeared in the article. This is because the results obtained by previous studies are more neutral, so the reference is cited. Here, the author agrees with the views and suggestions of the reviewers, so in this revision, the corresponding literature is replaced and modified to make it more in line with this article and increase the scientific nature of the article.] Thank you for pointing this out.